# Ocimum Sanctum [*Tulsi*]—An Alternative Additional Livelihood Option for the Poor and Smallholder Farmers

**Ghulam-Muhammad Shah** [1,*] **, Farid Ahmad** [1] **, Shailesh Panwar** [2] **, Manbar S. Khadka** [3] **, Ajaz Ali** [1] **and Suman Bisht** [1]

[1] International Centre for Integrated Mountain Development, Kathmandu 44700, Nepal;
farid.ahmad@icimod.org (F.A.); ajaz.ali@icimod.org (A.A.); suman.bisht@icimod.org (S.B.)

[2] Himalayan Action Research Centre, Dehradun 248006, India; shailesh@harcindia.org

[3] Former staff of International Centre for Integrated Mountain Development, Kathmandu 44700, Nepal;
skmanbar@gmail.com

\* Correspondence: ghulammuhammad.shah@icimod.org; Tel.: +977-1-527-5223

**Abstract:** The scope of *Tulsi* (*Ocimum sanctum*) as an alternative crop and means of livelihood, particularly for the rural poor, has not been extensively explored. *Tulsi* is in much demand for its medicinal and aromatic properties, particularly in Ayurveda. With increased depredation of cereal crops by wildlife, increased pest incidence, and decreasing availability of water for agriculture, an attempt was made to explore alternative livelihoods through *Tulsi* cultivation and value chain development. Using cross-sectional survey data of beneficiary households, the study employed the ordinary least squares method to examine the relationship between total crop income and the income from *Tulsi* for 2016 and 2017. The findings suggest that the average household's gross profit more than doubled within a span of two years. Total crop income of beneficiary farmers increased by 0.8 percent for every 1 percent increase in income from *Tulsi*. Most importantly, the intervention has shown a tremendous adoption rate. Initially, in 2013, 200 farmers cultivated *Tulsi* on 8.72 hectors of unirrigated and fallow land in the five beneficiary villages, but by 2017, towards the end of the intervention period, 400 farmers were cultivating the crop on 19.6 hectors of unirrigated, fallow land in 19 villages in Chamoli District. *Tulsi* farming and value chain development intervention not only provided marginal and smallholder farmers in these villages with a sustainable alternative additional livelihood option but also an opportunity where they were able to sustainably generate income from unirrigated, fallow land.

**Keywords:** *Ocimum sanctum*; alternative additional livelihood; crop depredation; out-scaling; *Tulsi*; value chain development

---

## 1. Introduction

There is growing consensus that sustainable farming is vital for reducing poverty in agriculture based economies [1]. Improving agriculture productivity of smallholder farming community is essential to realize the full potential of sustainable agriculture [2]. Common traditional crops generally grown in the hills and mountains of the countries located in the Hindu-Kush-Himalaya (HKH) region including wheat, barley, maize, potatoes, paddy, and millet which significantly contribute to the livelihoods as well as food requirements of these communities [3]. Agriculture in the mountains of the HKH region is generally constrained with limited land for agriculture, limited growing period, inadequate soil fecundity, deficient productivity and production, and inadequate post-harvest management impede attaining full potential of already limited agriculture resource base in these

regions [4]. Besides, in parts of the HKH region, for example, the hilly districts of Uttarakhand state in India, common traditional crops grown also suffer from depredation by wild animals, depleting water agriculture, and pest infestations. In the face of such vulnerabilities, provision of alternative but additional livelihood options for the poor and smallholder farmers becomes a high priority. At the same time mountain agroforestry products including non-timber forest products (NTFPs) and medicinal and aromatic plants (MAPs) are considered a substantial source of alternative livelihoods which could be treated and branded into mountain niche products also constrained with limited capacity to process and package these products and non-availability of established markets targeting such valuable products [5,6].

*Ocimum sanctum* also known as *Tulsi* or *Tulasi* in Sanskrit language and *holy basil* in English which belongs to *Lamiaceae* family of plants is one such NTFP also found in mountains and hills of the HKH region. *Ocimum sanctum*, an aromatic shrub, is a perennial plant with purple-pink flowers that produces light lemon scent. Historically, it is known for its healing properties that dates back over thousands of years. *Tulsi* is in much demand for its medicinal and aromatic properties, particularly in *Ayurveda*. The plant acts as a natural anti-stress agent and boosts immune system [7]. Besides, *Tulsi* is less water intensive crop and is less affected by animal depredation and pest-diseases as opposed to other major cereal crops [8]. *Ocimum sanctum* is also found in the semi-tropical and tropical parts of India where it is considered as an important medicinal plant [9]. Given its importance as medicinal and aromatic plant, *Ocimum sanctum* is also consumed as tea leaves. While *Tulsi* is believed to have originated in North Central India, it is widely popular today and is grown throughout the eastern world tropics [10]. About 50 million people in India alone count on NTFPs and MAPs as alternative livelihoods [11,12]. Data show that the collection and processing of medicinal and aromatic plants in India contribute at least 35 million working days of employment in a year [13]. Similarly, the global demand of MAPs has also increased substantially [14]. In fact, the global demand of medicinal and aromatic plants is growing at an annual rate of 5 to 15 percent. According to an estimate the world market for MAPs will reach 5 trillion USD by 2050 [15]. Ever changing agriculture practices, discoveries and innovations has significantly influenced the knowledge generation, sharing and use [16,17]. Agro-based industries promote agriculture value chains for enhancing, and sustaining agricultural growth. Such approaches also help generate sustainable employment opportunities in many poor and developing nations across the globe as they help to minimize the post-harvest losses and reduce rural poverty [18]. Therefore a value chain approach is considered as an effective way to improve horizontal and vertical linkages respectively between businesses and poor communities, and to develop enterprises targeting local valuable resources that not only benefit local communities but also help them sustainably tackle poverty [19,20]. Value chain development is comprised of a series of interconnected activities and facilitation where actors involved in the chain equitably befit in the process [21,22]. It also help generate employment at local level [23]. It is essentially related with the concept of upgrading products enabling actors involved to move up in the chain, add value to their products and services for optimum and equitable benefits [24]. Strengthening vertical and horizontal interaction at various stages of the value chain positively influence income of the stakeholders involved in the chain [25] and help reduce poverty among the beneficiary households [5]. Upgrading of products via value chain development entail attaining required skills and abilities in order to brand relevant products and services and leveraging niche markets for the same [26]. Essential actors in a value chain include producers of the product and services, collectors, traders, processors, and consumers who work together for the smooth functioning of the whole chain [27,28]. Value chain interventions depending on their nature and types tend to have positive impacts on the outcomes of interest. For instance, capacity building on improved harvesting practices not only enhances understanding of environmental values but also tends to raise bargaining and decision making skills thereby increasing the income of poor households involved in the value chain process [29]. The study of bay leaf value chains [30] also suggested that with an upgraded value chain, the bargaining power of rural households increases in terms of higher market prices for their produce. Many researchers have assessed value chain-based interventions and

their comparative impacts. For example, Choudhary et al., [31] study the prospects of developing a sustainable smallholder commercial production of African Leafy vegetables through pro-poor market development initiatives. They show that with the set-up of collective marketing systems and increased group efficiency through production skills training, poor and marginal farmers benefit with their increased participation in the market place [32]. Further, Cosyns et al. [33] analyze the impact of commercialization of medicinal and aromatic plants such as njansang (oily seed tree) on poverty alleviation in project villages of Cameroon. They find that project interventions help increase total cash income of the poor and marginal smallholder farmers in project households [32]. Similarly, Fischer and Qaim [34] investigate impacts of local cooperatives and farmer groups in engaging smallholder farmers objectively in banana value chains in Kenya. They find that while cooperative organizations may not necessarily improve market accessibility for smallholder farmers, the potential benefits to farmers are very product and context specific depending on the concrete and collective actions pursued. Bose [35] further argues that farmer groups have the potential to better link farmers with emerging high-value chains thereby increasing farmers' benefits and making the groups more sustainable. Along the similar lines, Krishna [36] assess the effect of engaging such smallholder groups at local level in Chamoli district of Uttarakhand, India. Their assessment argues that due to unorganized market mechanisms, producers sell their produce intermediaries at lower prices. They found that formation of smallholder farmer groups during value chain development process smallholder farmers become well organized, their bargaining power improves, and consequently sell their produce for a better price.

In essence, this paper argues that *Tulsi* farming provides a sustainable additional livelihood option and investigates impacts of *Tulsi* value chain development intervention on the beneficiary households as perused in the theory of change depicted in Figure 1 below. The paper also identifies factors contributing to increased income from *Tulsi* farming and assesses the contribution of *Tulsi* farming to household welfare, particularly benefits to women farmers.

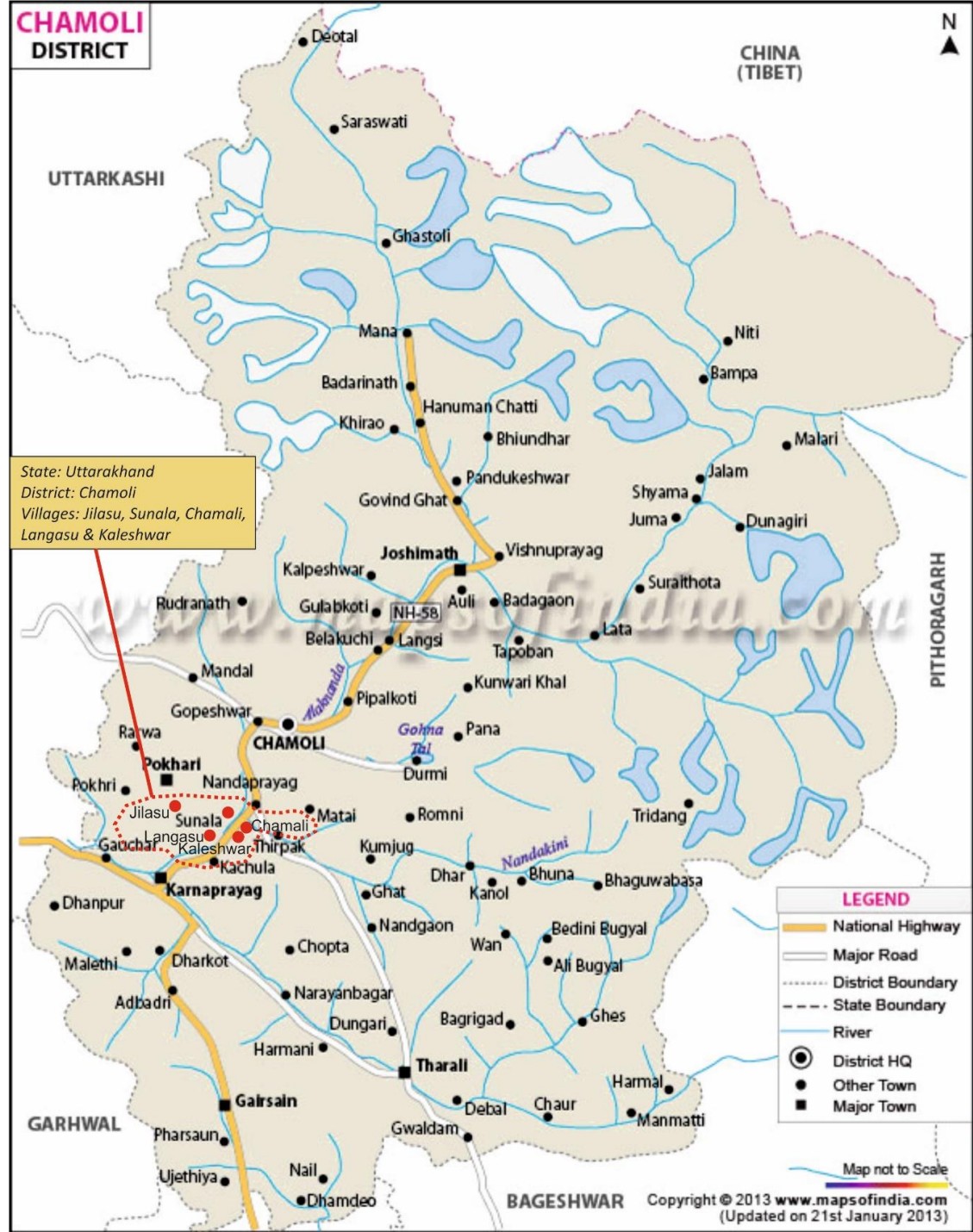

**Figure 1.** Map of the region showing the study areas and the location of the beneficiary villages.

## 2. Material and Methods

### 2.1. Study Area Context

Chamoli is one of the hilly districts of Uttarakhand state in India. The Karanprayag region of district Chamoli is rain fed with productivity of the food crops. The farmers and producers in the region are smallholder marginal farmers with less than 1 hectare of land. Outmigration of men for employment also means that there is shortage of farm labor in the area and females have to take the dual responsibilities of housework and work at their farms [37,38]. Besides, farmers in the

district have been facing numerous farm related problems such as crop depredation by wild animals, crop infestation by pests, and less availability of water for agriculture. Water scarcity is a major problem in the region. Farmers have reported that not only has the rainfall pattern changed (more intensive rainfall for shorter periods of time), but the water sources are drying up making irrigation and farming of crops difficult in general. Apart from low productivity, the farmers face severe problem of wild animals (wild boars and monkeys) damaging their crop and further reducing the food production ultimately adversely impact their annual income. Many farmers are forced to keep their farms fallow, if they are closer to the forest area, due to animal depredation thus further losing opportunity for food production. Sustainability of farming of traditional crops in these areas is challenged due to these reasons. However, replacing cultivation of traditional crops with any other farming practice is not only unsuitable but also irrelevant. More and more households are dependent on the market for purchase of staple crops. Alongside cultivation of traditional crops like wheat, barley, maize, potatoes, paddy, and millet, *Tulsi* has a good potential to provide an alternate additional livelihood option. More than 4.36% of land in the Chamoli district is barren and un-cultivated, and the Karanprayag region of the Chamoli district is rain fed with low productivity of food crops. Landscapes like these can be used for production of MAPs like *Ocimum sanctum* [37]. In this context, the *Tulsi* crop is considered as an important alternative additional cash crop for a number of reasons. First, it is less water intensive and is less affected by pest infestations, and there is no depredation of *Tulsi* crop by wild animals [8]. Second, *Tulsi* crop could be easily grown on unirrigated, barren, fallow, rain fed land. Third, value addition practices are comparatively easy to implement for enhancing the production of *Tulsi* via improvements along its value chain.

Farmers in Chamoli district have been practicing *Tulsi* cultivation in a traditional way with limited know-how on value addition. Realizing the potential of *Tulsi* as a viable alternative livelihood option and important alternative cash crop, the International Centre for Integrated Mountain Development (ICIMOD) initiated a value chain development intervention for *Tulsi* in Chamoli district of Uttarakhand state in India. As such *Tulsi* value chain development intervention strengthened farm-to-market nodes of *Tulsi* value chain bridging the gap between *Tulsi* producers and available market thereby enhancing the alternative livelihoods of poor and marginal smallholder farmers involved in *Tulsi* cultivation. *Tulsi*, value chain includes introduction of different varieties of *Tulsi*, improved and diversified production via innovation in existing supply chain. Introducing new technologies for drying and storage to minimize loss from wastage, streamlined processing, value additions through development of new by-products, attractive packaging, formation of cultivator's groups, strengthening of institutional and marketing management systems of the cooperatives as well as identification of market opportunities and strengthened market linkages by expanding retail and distribution networks of *Tulsi* by-products produced and marketed by women-led cooperatives for better returns and sustainability of the value chain.

### 2.2. Description of Tulsi Value Chain Development Intervention

*Tulsi* value chain development intervention was implemented in five villages in Chamoli district of Uttarakhand state in India. ICIMOD implemented this intervention through its local partner Himalayan Action Research Centre (HARC), Dehradun-Uttarakhand, India. Intervention was supported through a grant funded by International Fund for Agriculture Development (IFAD). The intervention was implemented over a period of five years, 2013–2017 targeting mountain farmers with small farm holdings and who were vulnerable to crop damages from wild animals' especially wild boars and monkeys and lack of water due to climate stress. Through a preliminary situational analysis *Tulsi* was identified as a crop that could reduce these vulnerabilities as such. Part of the assessment, alongside cultivation of traditional crops, willingness of farmers to also cultivate *Tulsi* as an alternative additional livelihood option was assessed. Beneficiary farmers were selected based on their interest to participate in the intervention. The intervention initially supported 200 farmers in 5 villages namely Jilashu; Sunala; Chamali; Langashu; and, Kaleswhar in Chamoli district. Among the

beneficiaries 90% were women farmers. Beneficiary farmers were small and marginal farmers with average land holdings between 0.70 and 1.34 hectares of land. Beneficiary farmer cultivated *Tulsi* on a total of 8.72 hectors of unirrigated, barren/fallow land in these 5 villages.

The main objective of this intervention was to provide a sustainable alternative additional livelihood option to the beneficiary households by providing necessary training, technology, and market linkage support. It aimed to do so by strengthening *Tulsi* value chain coordination, improving processing and functional up gradation and, diversifying *Tulsi* products. For this purpose, intervention adopted a community led integrated approach to address complex issues of diversification of farm production and introduced efficient marketing strategies. Farmers were mobilized and briefed about potential of *Tulsi* farming for additional income generation while continuing cultivation of traditional crops. Part of capacity building support, information and techniques on nursery management, sowing methods, quality harvesting and post-harvest handling were transferred to farmers. A comprehensive package of practice on *Tulsi* farming and production was developed and shared across the beneficiary households. In order to address issues related to marketing of *Tulsi* products a market surveys was carried out in surrounding markets of Uttarakhand and in the national capital region (NCR) in Delhi. Findings suggested that *Tulsi* leaf in its different forms are in demand by consumers. Based on these findings, product development trails were taken up by the intervention. *Tulsi* green tea, *Tulsi* ginger tea, *Tulsi* powder, and *Tulsi* sauce were introduced. Intervention helped beneficiary communities to assemble in *Tulsi* producers' and collectors' groups. These groups were further trained in managing newly introduced *Tulsi* products. This in return not only provided poor farmers an alternative livelihood option but also helped value chain governance in terms of enhanced coordination of *Tulsi* production and local level trade which also resulted in transparency and equity in *Tulsi* value chain.

Figure 2 below depicts the theory of change (ToC) realized for *Tulsi* value chain development intervention. ToC was developed using 'Participatory Impact Pathways Approach' when the study team was designing this study. Major stakeholders of the intervention including vale chain development team, representatives from the implementing organization, and village head of the beneficiary villages participated in this discussion. The following questions helped study team to prepare the overall ToC pursued for this intervention.

Q1. What were the desired impacts expected from this intervention? Consolidated response(s) are shown in the last right hand side column in Figure 2;
Q2. What were the immediate, intermediate, and higher level results expected from this intervention? Consolidated responses are shown in the middle columns 2, 3, and 4 in Figure 2; and,
Q3. What the intervention has done differently in order to achieve desired, ultimate results of the intervention? Consolidated responses are shown in the 1st column in Figure 2.

Intervention theory of change suggests that important intervening factors not only helped beneficiary farmers realize increased proportion of sale volume of *Tulsi* produce, but also enable them to negotiate a better price for their produce. These factors include handholding support provided to beneficiary farmers through establishment of women self-help groups of *Tulsi* producers and collectors and capacity building of farmers in *Tulsi* nursery establishment. It also includes management, harvesting and quality post-harvest handling—collecting, drying, grading, and packaging of *Tulsi* in addition to linking them to the market. Capacity building and strengthening of coordination between groups of *Tulsi* producers, collectors, and buyers led to smooth flow of upstream, downstream information and trade thereby improving governance, transparency and equity in *Tulsi* value chain. These intermediary factors collectively led to increased farm-gate price received by *Tulsi* producers, increased benefits to women beneficiaries, and increased social benefits and well-being of poor and marginal farmers. The intermediary and higher level outcomes farmers in Chamoli district to adopt *Tulsi* as an alternative livelihood option leading to increased income of beneficiary households from

*Tulsi* farming. Findings of the study discussed in the results section of this paper also suggest that the theory of change realized for this intervention remained valid.

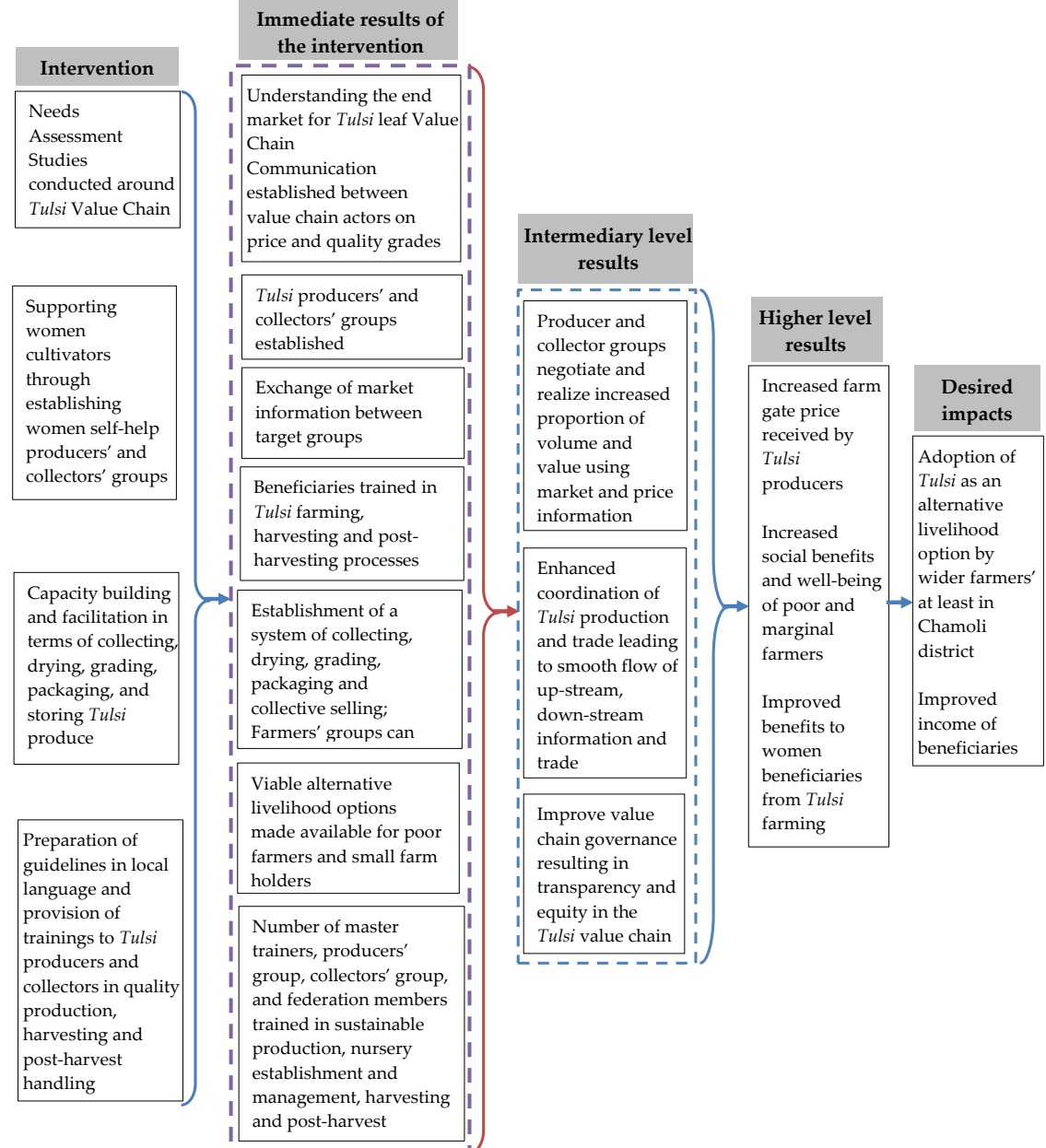

**Figure 2.** Theory of Change realized for *Tulsi* value chain development intervention.

### 2.3. Empirical Approach

A cross-sectional survey was conducted involving beneficiary households in five of the beneficiary villages using a household survey questionnaire. Household questionnaire for this study was developed following the Theory of Change logic. Indicators relevant to intermediary, higher level results and desired impacts were included in the household survey questionnaire. Survey questionnaire was pilot tested and finalized for formal data collection. Given the fact the overall objective of the intervention was to provide an alternative additional livelihood option to the beneficiaries and to increase their income from *Tulsi* value chain development intervention, the empirical approach and the regression model used in this study mainly considered income indicators. Aiming at further validating quantitative findings of the study, focus group, and key informant discussions were conducted with

beneficiary households. The ordinary least squares (OLS) method was used to examine the relationship between total crop income and the income from *Tulsi* for the years 2016 and 2017 while controlling for other explanatory variables such as total value of crop consumption (excluding *Tulsi*), total value of crop production (excluding *Tulsi*), *Tulsi* production expenses, and value of total crop damages from various threats in 2016 and 2017. The findings obtained from quantitative survey were further triangulated and supplemented from the qualitative survey data gathered from focus group, and key informants' interviews. Given the consideration that data was collected through a cross-sectional survey, we preferred to use OLS method.

The model used in this paper is:

$$Ln\,(Y) = \alpha + \beta_1\,Ln\,_{(x_1)} + \beta_2 Ln\,_{(x_2)} + \beta_3\,Ln\,_{(x_3)} + \beta_4\,Ln\,_{(x_4)} + \beta_5\,Ln\,_{(x_5)} + \varepsilon \tag{1}$$

where, $Ln(Y)$ is the dependent variable, known as log of total crop income in 2017 and is the function of $Ln\,_{(x_1)}$ (Log of total value of crop consumption in 2017), $Ln\,_{(x_2)}$ (Log total value of crop production in 2017), $Ln\,_{(x_3)}$ (Log of *Tulsi* income in 2017), $Ln\,_{(x_4)}$ (Log of *Tulsi* production expenses in 2017), $Ln\,_{(x_5)}$ (Log of total value of crop damage in 2017), and $\varepsilon$ (Error term) and $\alpha$ (Constant). $\beta_i$ represents regression coefficient of these explanatory variables.

Similar model has been applied to analyze the relationship between total crop income and total *Tulsi* income in 2016. This model assumes that total crop income will have positive relationship with *Tulsi* income, total value of other crop production but negative relationship with the value of crop damages, *Tulsi* production expenses and total value of crop consumption. In addition to linear regressions, descriptive statistics, cross tabulations, and specific figures have been used to identify the changes in cropping patterns, possible major threats to cereal crops, and farmers' coping strategies, among others.

## 3. Results

### 3.1. Adoption of Tulsi over the Years

Data collected from focus group, and key informant discussions suggest that the intervention has shown a tremendous adoption rate (Table 1). Intervention was started in 5 villages where total number of *Tulsi* cultivators was found to be 200 farmers. Beneficiary farmer cultivated *Tulsi* on a total of 8.72 hectors of unirrigated, barren/fallow land in these 5 villages. Towards the end of the intervention, HARC out-scaled value chain work to 19 additional villages in Chamoli District, thereby reaching out to more than 400 additional households who were cultivating *Tulsi* on a total of 19.6 hectors of unirrigated, barren/fallow land across 19 villages. This suggests that *Tulsi* farming and value chain development intervention not only provided marginal and smallholder farmers in these villages with a sustainable alternative additional livelihood option but also an opportunity where they were able to sustainably generate income from unirrigated, fallow land. Trend showing promising adoption suggests that *Tulsi* crop has a good potential for being scaled-out to other villages, including neighboring districts as a sustainable alternative additional livelihood option.

### 3.2. Benefits to Women Beneficiaries

Given the involvement of women in agriculture, one can expect the potential benefits to female household members from *Tulsi* cultivation. Respondents were asked specifically about benefits of *Tulsi* farming to women beneficiaries. Findings suggests that the collection and cultivation of *Tulsi* has provided an important source of sustainable alternative additional cash income to rural communities especially for women. Assessment findings further suggest that the intervention enabled 97% of the female beneficiaries earn an independent income from *Tulsi* farming. In terms of involvement of women at household level decision making, nearly all the respondents said that routine decisions pertaining to the household were made jointly by male and female heads. The cash income was usually

kept by both male and female heads of the household and all the respondents said that major decisions on household expenditure were made jointly by both male and female heads of the household.

**Table 1.** Adoption rate of *Tulsi* farming observed over the years.

| Year | Number of Villages | Number of Cultivators | Total Agricultural Land (in Hectors) | *Tulsi* Cultivation in Un-Irrigated Land (Hectors) | *Tulsi* Production on Barren, Fallow Land (Hectors) | Total Volume (Tons) | Total Value (100,000—Indian Rupee, INR) |
|---|---|---|---|---|---|---|---|
| 2013 | 5 | 200 | 203.48 | 7.48 | 1.24 | 87 | 6.10 (USD. 8288) |
| 2014 | 8 | 211 | 274 | 8.44 | 2.10 | 126 | 10.12 (USD. 13,750) |
| 2015 | 10 | 234 | 297 | 9.16 | 2.26 | 140 | 11.23 (USD. 15,258.2) |
| 2016 | 11 | 259 | 312 | 10.16 | 2.50 | 155 | 12.43 (USD. 16,888.6) |
| 2017 | 19 | 400 | 488 | 16.12 | 3.40 | 241 | 19.28 (USD. 26,195.7) |

Note: The monetary values are reported in Indian currency denoted as INR. And 1 USD (US dollar) is equivalent to INR 73.6 in 2018.

### 3.3. Threats to Crop Cultivation

The major crops like paddy, wheat and others have suffered from depredation by wild animals as well as pest infestations. The respondents were asked about the three most critical threats that have affected their crop cultivation in recent years. Notably all the farmers reported crop depredation by wild animals, water scarcity for cultivation and pest diseases as the three most frequent threats to farm cultivation.

In response to such threats, farmers were asked about the three most preferred alternative crops that they have been growing in recent years. Interestingly, 100 percent of the respondents said that they grew *Tulsi* as the first most preferred alternative to major crops (Figure 3). This shows that in the face of such vulnerabilities, *Tulsi* farming has gained prominence among the farmers in the intervention areas as a sustainable alternative source of earning an additional income. More than three quarters of the respondents said that they grew dal (dal also written as dall, is a term used in the Indian subcontinent for dried, split pulses including lentils, peas, and beans) as the second most preferred alternative to major crops. *Tulsi* farming requires far less water for cultivation as opposed to paddy cultivation and other major cereals. This also suggests that *Tulsi* farming has potential to increase resilience of farmers in responding to such threats and vulnerabilities. This does not mean that beneficiary farmers have abandoned cultivating other common crops but *Tulsi* cultivation has provided farmers with an alternative mean of earning a sustainable additional income. Figure 3 clearly suggest that figure millet and paddy are still the 2nd and 3rd most preferable crop beneficiary farmers have been cultivating over the years.

### 3.4. Factors Influencing Earning from Tulsi Farming

Given the fact that *Tulsi* farming has gained prominence among the farmers in the intervention areas. Farmers were asked about the various factors that have enabled them to earn more from *Tulsi* cultivation. Almost 94 percent of the respondents said that the first main factor that enabled them to maximize earnings from *Tulsi* farming was the formation of *Tulsi* producers and collectors groups. Finding of key informant discussions further revealed that formation of these groups not only helped beneficiaries realize increased proportion of sale volume of *Tulsi* produce but also enable them to negotiate better price for their produce. Nearly 89 percent of the respondents said that the second main factor that enabled them to earn more from *Tulsi* farming was capacity building which helped them better manage *Tulsi* farming in terms of nursery establishment and management,

quality harvesting and post-harvest handling of *Tulsi* produce. Slightly more than 90 percent of the respondents said that the third main factor that enabled them to maximize earnings from *Tulsi* farming was the increased productivity.

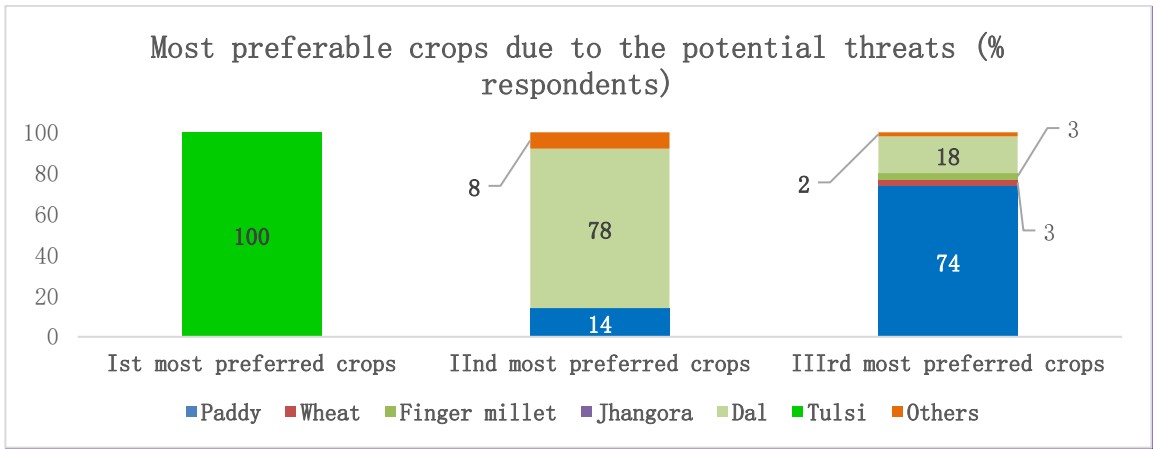

**Figure 3.** Most preferable crops due to the potential threats (percent of respondents).

### 3.5. Other Benefits of Tusli Farming

The respondents were also asked about the specific ways in which *Tulsi* farming had benefitted the rural households. Two-thirds of the respondents said that their household income had increased as a result of *Tulsi* farming. Slightly more than one-tenth of the respondents said that they have been able to meet daily family needs with increased earnings from *Tulsi* farming (Table 2).

**Table 2.** Other benefits from Tulsi farming.

| Tulsi Farming Contribution | Percent of Responses |
|---|---|
| Increment in income source | 68 |
| Meet daily family needs | 14 |
| Increase saving | 3 |
| Increase purchasing power | 3 |
| Others | 12 |

### 3.6. Improved Income and Profits from Tulsi Farming

Relationship between the total income from crops and from the sale of *Tulsi* was calculated using ordinary least square regressions method for two different time periods 2016 and 2017. Table 3 provides a summary statistics of key variables used in this empirical analysis. Worth mentioning here is the finding that the mean profit from the sale of *Tulsi* has substantially increased from INR. 2631 in 2016 to INR. 5478 in 2017 (Table 3). At the same time, the average gross profit from total crop farming also increased from a loss of INR. 1622 in 2016 to INR. 942 in 2017. Findings from the key informant discussions with beneficiary households suggest that increase in profit from sale of *Tulsi* is because formation of *Tulsi* producers and collectors groups, and their capacity building, which has benefited beneficiary farmers.

Analysis of the OLS regression results including the correlation between total crop income and income from *Tulsi* in 2016 and 2017 respectively has been shown in Table 4 below. In the first model, the dependent variable is the log of total crop income in 2017. The explanatory variables are log of total value of crop consumption in 2017, log of total value of crop production in 2017, log of *Tulsi* income in 2017, log of *Tulsi* production expenses in 2017 and log of total value of crop damage in 2017. Interestingly, as illustrated in Table 4, we find that the total crop income for farmers increases by 0.8

percent for every 1 percent increase in *Tulsi* income in 2017. This relationship is statistically significant at 1 percent level. The findings of this study are consistent with the study findings conducted by [5,32–34] that analyze the impact of commercialization of medicinal and aromatic plants such as njansang on poverty alleviation in project villages of Cameroon.

**Table 3.** Summary Statistics.

| Variables | Mean | Standard Deviation | Minimum | Maximum |
|---|---|---|---|---|
| Profit from *Tulsi* in 2016 | 2631 | 1994 | 600 | 8400 |
| Value of total crop consumption in 2016 | 3173 | 9796 | 0 | 78,750 |
| Value of total crop production in 2016 | 3068 | 3371 | 0 | 12,000 |
| *Tulsi* income in 2016 | 3210 | 2169 | 1050 | 9450 |
| *Tulsi* production expenses in 2016 | 580 | 200 | 320 | 1400 |
| Value of total crop damage in 2016 | 98 | 13 | 0 | 103 |
| Profit from *Tulsi* in 2017 | 5478 | 9405 | 705 | 53,260 |
| Value of total crop consumption in 2017 | 2166 | 4193 | 0 | 30,400 |
| Value of total crop production in 2017 | 3600 | 7981 | 0 | 60,800 |
| *Tulsi* income in 2017 | 6103 | 9448 | 1155 | 53,960 |
| *Tulsi* production expenses in 2017 | 625 | 216 | 250 | 1400 |
| Value of total crop damage in 2017 | 98 | 12 | 0 | 100 |
| Total crop profit in 2017 | 942 | 8625 | −5975 | 50,730 |
| Total crop profit in 2017 | −1622 | 2533 | −6250 | 6050 |

Note: the monetary values are reported in Indian currency denoted as INR. And 1 USD (US dollar) is equivalent to INR. 73.6 in 2018.

**Table 4.** OLS regression estimates showing the correlation between total crop income and income from *Tulsi* respectively in 2016 and 2017.

| (1) | (2) | (3) |
|---|---|---|
| | **Log Total Crop** | |
| **Variables** | **Income in 2017** | **Income in 2016** |
| Log (Total Value of Crop Consumption in 2017) | −0.5 * (0.3) | – |
| Log (Total Value of Crop Production in 2017) | 0.9 * (0.3) | – |
| Log (Tulsi Income in 2017) | 0.8 *** (0.1) | – |
| Log (Tulsi Production Expenses in 2017) | −0.2 (0.2) | – |
| Log (Total Value of Crop Damage in 2017) | 5.5 (5.4) | – |
| Log (Total Value of Crop Consumption in 2016) | – | 0.0 (0.1) |
| Log (Total Value of Crop Production in 2016) | – | 0.3 *** (0.1) |
| Log (Tulsi Income in 2016) | – | 0.9 *** (0.1) |
| Log (Tulsi Production Expenses in 2016) | – | −0.4 ** (0.1) |
| Log (Total Value of Crop Damage in 2016) | – | 0.1 (0.1) |
| Constant | −24.9 (25.1) | 0.7 (0.7) |
| Observations | 39 | 44 |
| R-squared | 0.9 | 0.9 |

Robust standard errors in parentheses. *** denote significance at 1 percent level, ** denote significance at 5 percent level and * denote significance at 10 percent level.

Similarly, the farmers' total crop income increases by 0.9 percent for every 1 percent increase in total value of crop production in 2017. This relationship is statistically significant at 1 percent level. In the second model (column 3, Table 4), we find that the total crop income for farmers increases by 0.9 percent for every 1 percent increase in *Tulsi* income in 2016, and this relationship is statistically significant at 1 percent level.

### 3.7. Rate of Return from Tulsi Farming

Interestingly, the study finds that the rate of return from *Tulsi* farming is higher than that of cereal crop farming. Analysis of profit and loss from *Tusli* farming versus cereal crops suggests that the latter take at least six to seven months from sowing to harvesting while basil crops take only three months. This means that within a six month period (May–October), two cycles of *Tulsi* crop can be harvested. This is an indication that while sustainability of farming of traditional crops in these areas is challenged, *Tulsi* cultivation provides a sustainable alternative additional income opportunities across its value chain such as leaf plucking, drying, blending, and packaging (Table 5).

**Table 5.** Rate of return analysis for the cultivation of *Tulsi* versus cereal crops (paddy).

| Description | Paddy | | Tulsi | |
|---|---|---|---|---|
| | Time/Quantity | Cost (INR) | Time/Quantity | Cost (INR) |
| Seed (grams) | 2000 g | 50 | 50 g | 50 |
| Nursery preparation/seed sowing (day) | 1 | 200 | $\frac{1}{2}$ | 100 |
| Irrigation in nursery (days) | 1 | 200 | $\frac{1}{2}$ | 100 |
| Weeding & Hoeing (day) | 1 | 200 | 1 | 200 |
| Ploughing (day) | 1 | 200 | 1 | 200 |
| Manuring (day) | $\frac{1}{2}$ | 100 | $\frac{1}{2}$ | 100 |
| Transplanting (day) | 1 | 200 | $\frac{1}{2}$ | 100 |
| Irrigation in filed (day) | 1 | 200 | $\frac{1}{2}$ | 100 |
| Weeding & Hoeing of field (day) | 1 | 200 | 1 | 200 |
| Harvesting (day) | $\frac{1}{2}$ | 100 | $\frac{1}{2}$ | 100 |
| Transportation (day) | $\frac{1}{2}$ | 100 | $\frac{1}{2}$ | 100 |
| Threshing/Sorting | 1 | 200 | $\frac{1}{2}$ | 100 |
| Drying/Packing/Storage | 1 | 200 | $\frac{1}{2}$ | 100 |
| Total cost (INR) | | 2150 | | 1550 |
| Output & value (in Kg) | 50 | 750 | 400 | 3600 |
| Net profit & loss (0.02 hectare land) | | −1400 | | 2050 |
| Crop cycle from nursery to harvesting (months) | 6–8 | | 3–4 | |

Note: the comparison is based using an equal availability of arable land i.e., 0.02 hectare for Tulsi and paddy cultivation. The selling price of paddy is INR 15 per kg while that of green leaves is INR. 9 per kg. The monetary values are reported in Indian currency denoted as INR. And 1 USD (US dollar) is equivalent to INR. 73.6 in 2018.

It is also clearly indicative from this analysis that farmers incur a net loss of INR. 1400 with the cultivation of paddy in 0.02 hectares of land from May through October, whereas they gain a net profit of INR. 2050 with the cultivation of *Tulsi* in the same proportion of land within a period of three to four months (Table 5). Therefore, *Tulsi* provides a clear benefit to smallholder farmers as opposed to paddy which in return could increase wellbeing of poor and marginal farmers.

## 4. Conclusions, Discussion, and Way Forward

*Tulsi* value chain development intervention has created pursued impacts at the household level. *Tulsi* farming and value chain development intervention not only provided marginal and smallholder farmers in the beneficiary villages with a sustainable alternative additional livelihood option but also an opportunity where they were able to sustainably generate income from unirrigated, fallow land. However, it is highly desirable that such interventions recognize and address gender and social equity aspects particularly pertaining to women, so that women are not only able to earn an independent income but their social equity is also recognized. The agricultural requirement for cultivating *Tulsi*

is found to be less as compared to other traditional crops grown in Chamoli district. On the other hand *Tulsi* is less water intensive, and less affected by pests and animal depredation, thereby reducing risks of poor and marginal farmers to these vulnerabilities. This also suggests that *Tulsi* farming has potential to increase resilience of farmers in order to respond to threats and vulnerabilities to other traditional crops grown in the district. Given the fact the overall objective of the intervention was to provide an alternative additional livelihood option to the beneficiaries and to increase household level income of the beneficiary households from *Tulsi* value chain development intervention, the empirical approach used in this study mainly considered income indicators. Data was collected at the end of the intervention during 2017 through a cross-sectional survey of beneficiary households in five of the beneficiary villages. Household level baseline data was not available and data for the year 2016 used in this study was collected using recall method which has its own limitations for being considered as panel data. Assessing efficiency of the intervention using a cost-benefit approach is essential, remains an area to explore further.

The intervention applied a holistic approach to address complex issues of diversification of farm production and introduction of efficient marketing strategies with local institution. It facilitated capacity building of beneficiaries and social mobilization through the formation of *Tulsi* producers' and collectors' linking them to available markets. Hence, the integrated and community led value chain approach adopted by the intervention has proved to be the basis for generating additional income from *Tulsi* thereby improving their income resilience. Therefore, given the findings of this study, *Tulsi* value chain as an alternative additional source of livelihood diversification option could be further expanded enabling marginal and smallholder farmers earning a sustainable alternative additional income. This could be out-scaled at district level in the larger Chamoli district in particular and up-scaled to other districts of Uttarakhand state in India. However, this will require commitment and support from the government as well as development agencies. Similarly, expanding the scale of product testing and diversifying *Tulsi* products requires equal commitment and policy support from government.

Market access and linking farmers to available markets is among the key drivers for adopting *Tulsi* as an alternative additional crop. Farmers adopt alternative crops if their risks from market failures and loss of income are adequately addressed. Therefore, reducing risk and vulnerability of farmers and processors against market failure is required to be seen as an integral part of adaptation strategies. This could be attained through further strengthening market linkages with producers, buyers and end users for producing and marketing *Tulsi* products. Besides, process and functional upgrading for different value added products from *Tulsi*, for example, *Tulsi oil, Tulsi powder, Tulsi Tea* and niche branding, quality production, and identifying relevant consumer market for products could have positive and multiplier effects on beneficiaries' income. This would require establishing *Tulsi* as a niche product at least in the national market. At the producers' level, this would require further capacity building in post-harvest practices like proper drying, quality grading and storage. At the processors level this would require introduction of new technology for quality processing and packaging. At the same time, strengthening of community-led local institutions in processing and marketing of *Tulsi* products will be required.

**Author Contributions:** Conceptualization, G.-M.S. and F.A.; Methodology, G.-M.S., F.A. and M.S.K.; software, G.-M.S. and S.P.; validation, S.P. and G.-M.S.; formal analysis, M.S.K. and G.-M.S.; investigation, G.-M.S.; resources, S.B. and S.P.; data curation, S.P. and G.-M.S.; writin—original draft preparation, M.S.K. and G.-M.S.; writing—review and editing, A.A. and S.B.; visualization, G.-M.S. and A.A.; Supervision, F.A.; project administration, S.B. and S.P.; funding acquisition, ICIMOD.

**Funding:** This evaluation research, and APC was funded by [ICIMOD core funds] and the intervention was funded by [IFAD].

**Acknowledgments:** The authors would like to thank the anonymous reviewers and editors of this paper. We would also like to acknowledge support of the International Fund for Agriculture Development (IFAD) for supporting this intervention. The authors gratefully acknowledge the support of core donors of ICIMOD: the Governments of Afghanistan, Australia, Austria, Bangladesh, Bhutan, China, India, Myanmar, Nepal, Norway, Pakistan, Switzerland, and the United Kingdom. The views and interpretations expressed in this paper are those of the authors and are not attributable to ICIMOD or any other organizations.

**Conflicts of Interest:** The authors of this article declare no conflict of interest.

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
