# Peer review of "Ocimum Sanctum [Tulsi]—An Alternative Additional Livelihood Option for the Poor and Smallholder Farmers"

_sustainability, doi:10.3390/su11010227_

Reviewer 1 Report

This manuscript describes an interesting story whereby the livelihoods of farmers in Uttarakhand State, India are improved by converting production away from core crops such as wheat, barley and maize to a niche crop known as Tulsi whilst simultaneously having access to various support mechanisms to improve education, markets and capacity building associated with the Tulsi value chain. The outcome is a doubling of household livelihoods of two years. There is no doubt to me that the story within the manuscript is interesting and the impact on livelihoods positive but I have a number of quite serious concerns regarding how the study is reported. 

 Firstly, the journal is entitled ‘Sustainability’ but the content of the manuscript does not match fully that of the journal as there is no evidence presented that the change to the niche crop is indeed sustainable. Yes social and economic aspects of the individual farms are the focus and are encouraging but environmental aspects are largely glossed over. At lines 128-129 and 334-337 the text mentions that the Tulsi crop has less input demands and is less prone to pest and disease problems but no evidence what so ever is given to support these claims. No other evidence is given to show that converting to Tulsi production is in fact a sustainable choice. This aspect needs to be strengthened considerably with sound evidence.

In addition and closely related at line 255 and in the Abstract the script mentions that more and more farmers are converting to growing Tulsi but there is no discussion of longer term issues that might arise if, for example, core traditional crops are not grown as much - what are the potential issues for householders? Will they still have ample access to traditional crops? What are the consequences if they do not? Are there potential nutritional issues? Will it force up prices of traditional crops and disturb markets? What if the market becomes swamped with Tulsi and the market value drops?  The manuscript needs to discuss the broader sustainable implications of mass swapping from growing a traditional staple crop to a niche crop. There are currently too many unanswered questions regarding the sustainability of the Tulsi intervention.

 Secondly, aside from the financial aspects, most of the other benefits mentioned on the manuscript such as the ‘benefits to women’ and ‘threats to crop cultivation’ are not verifiable from the data provided so have no real scientific credibility and only have anecdotal value. These aspects of the manuscript should be supported by data or removed. However, removing them will also remove the social aspect of sustainability and so less match the journal objectives. This aspect is partly linked to a poor methodological description of the study and the research question and, in particular, a lack of information on how data has been collected and analysed.Currently, the manuscript is written more like a report than that of a scientific publication. In any scientific manuscript the results/conclusions must be verifiable from the raw data. This is not the case here. 

Some examples of where greater clarity is needed:

- The time scale of the study is not actually clear. Data is presented for 2016 and 2017 but both these years seem to be for Tulsi production. There does not seem to be a baseline (unless I have missed it somehow) of data for pre-Tulsi production. This is critical otherwise the there is no evidence what so ever that Tulsi production is actually an improvement on the pre-Tulsi situation.

- In section 2.2 for example the manuscript mentions a survey but very little information is given. For example, why were only 5 villages surveyed? What is meant by a ‘cross-sectional survey’ in this context? How many households in each beneficiary village were surveyed? How were these households selected? What type of questions were asked? How was the data recorded and analysed? How was missing data handled? Indeed I would expect to see the survey provided as supplementary data. 

- At line 201 the text mentions ‘selected stakeholders’ please clarify - selected by whom? What were the selection criteria?

 Use of the English language is not poor by any standards but the manuscript does lack clarity and readability in many places and the main reason for this is an excess of short sentences that read rather repetitively. That aside the content of the introduction is quite good and sets the scene well. It is also quite well referenced.  However, I would recommend that if the manuscript is re-submitted then it should be proof-read and corrected by a native English speaker or a professional editing service used.

 Other issues:

- The abstract should briefly explain what Tulsi is, many people will not know the top. The abstract must also briefly mention the methodological approach.

- A map of the region in section 2.showing the study areas and the location of the beneficiary villages would be useful.

- A clear, concise explanation of exactly what is meant by ‘Tulsi value-chain intervention’ is required early on in the manuscript. 

Author Response

Thank you for your highly constructive feedback. Your feedback has helped us further refine and improve our manuscript. We hope that we are able to appropriately incorporate/report your feedback to our best given design of the study and limitations. We are attaching herewith our responses for your consideration please. 

Reviewer 2 Report

Title:

Introduce in the title the Latin name for Tulsi plant;

Abstract:

is not self-explanatory. Provide more information about the scope of the study, methods used and main results;

Introduction:

use italic for the Latin names (Ocimum sanctum, etc); Objectives of the study: why these objectives? What does this paper bring new? Why not assessing the efficiency of the intervention programme (in a cost –benefit approach) to provide answers to different stake-holders for other possible programmes designed to support the self-sufficiency of small holders?

Material and methods:

- Description of Tulsi value chain development intervention: what was the cost of the programme? Who supported this; How many women between the farmers? It is not clear the link between the figure 1 mentioned in line 173 and the programme.

- Empirical approach. More information about questionnaire designed and the survey techniques are needed. What are the limits of model 1? Better explain why using that variables.

Results:

- Descriptive statistic for the data is missing; Transform all data related to income and costs in USD; Is the programme efficient? Are the households self-sufficient?

Discussion:

- Discussion is missing from the paper. In which conditions such a programme can be multiplied elsewhere?

Author Response

Thank you for your highly constructive feedback. Your feedback has helped us further refine and improve our manuscript. We hope that we are able to appropriately incorporate/report your feedback to our best given design of the study and limitations. Attached please find our responses for your consideration please. 

Reviewer 3 Report

The methodology of the empirical was mentioned clearly, the source of data of the figures are the personal effort of the authors.

Author Response

Thank you for appreciating study methodology and the empirical model used in this study.  Your feedback on the figures has helped us further refine the write-up pertaining to the figures and improve our manuscript. We hope that we are able to appropriately incorporate/report your feedback to our best given the design and limitations of the study. Attached please find our responses for your consideration please. 

Reviewer 4 Report

The authors assessed the impacts of Tulsi value chain development on the livelihoods of rural poor in India.

 A cross-sectional survey was carried out using a household survey questionnaire.

The OLS method was used to examine the relationship between total crop income and the income from Tulsi during the period 2016-2017, in addition to other explanatory variables, i.e. total value of crop consumption (excluding Tulsi), total value of crop production (excluding Tulsi), Tulsi production expenses, value of total crop damages from various threats.

 The findings obtained from the quantitative survey were integrated by qualitative survey data gathered from key informants’ interviews.

 Results highlighted that the average households’ gross profit from Tulsi farming increases by more than double. Besides, total crop income of beneficiary farmers’ increases by 0.8 percent for every 1 percent increase in Tulsi income.

The study also found that: the rate of return from Tulsi farming is higher than that of cereal crop farming; the collection and cultivation of Tulsi provide an important source of cash income to rural communities, especially for women; Tulsi is the first most preferred alternative to major crops to face crop depredation by wild animals, water scarcity for cultivation and pest diseases; the main factors that enabled farmers to maximize earnings from Tulsi farming are: 1) the formation of Tulsi producers and collectors groups, 2) the capacity building which helped them a better management of Tulsi farming in terms of nursery establishment and management, quality harvesting and post-harvest handling, 3) the increased productivity.

 The study is original for the plant species and the geographic area considered, as well as in terms of practical results from the implemented intervention for the Tulsi value chain development.

However, in order to make the paper publishable, I think that the authors should clarify the following aspects in the text:

 a) Is the OLS method a suitable approach for analysing the gathered data, or could a panel data analysis give a better interpretation of results? If so, a new analysis should be carried out.

 b) Are the independent variable affected by multicollinearity? In this case, a better selection of predictors should be carried out and a new OLS/panel data analysis should be implemented.

 Besides, I think that the sections 3.3, 3.4, 3.5 and 3.6 should precede the OLS results, and that the regression results should be further commented based on the considered regressors.

Author Response

Thank you for your highly constructive feedback. Your feedback has helped us further refine and improve our manuscript. We hope that we are able to appropriately incorporate/report your feedback to our best given the design and limitations of the study. Please find attached herewith our responses for your consideration please.  

Round  2

Reviewer 1 Report

This manuscript describes an interesting story whereby the livelihoods of farmers in Uttarakhand State, India are improved by converting production away from core crops such as wheat, barley and maize to a niche crop, Ocimum danctum, known as Tulsi, whilst simultaneously having access to various support mechanisms to improve education, markets and capacity building associated with the Tulsi value chain. The outcome is a doubling of household livelihoods of two years.

This is the second time I have reviewed this manuscript. Initially I had major concerns as to how the study was reported despite it being an interesting study.

This version of the manuscript is much improved. However there is still a few important aspects that have not been considered which are of vital importance if the approach is to improve agricultural sustainability in the future and that is whether of not, taking a longer viewpoint and considering environmental impact from reducing the diversity of crops grown in the area, converting to Tulsi production is in fact a sustainable choice. This aspect must be discussed.

Use of the English language much better but still has a few issues. I would recommend that the manuscript should be proof-read and corrected by a native English speaker or a professional editing service used.

Author Response

Thank you for the generosity of your time reviewing our manuscript the 2nd time. Please, kindly find attached our responses.

Reviewer 2 Report

The authors improved the paper accordingly to the suggestions/comments pointed out in the first review process. I recommend to publish the paper.

Author Response

The authors improved the paper accordingly to the suggestions/comments pointed out in the first review process. I recommend to publish the paper.

Thank you for generosity of your time reviewing our manuscript 2nd time. We are pleased to see that we were able to incorporate your feedback appropriately in the revised manuscript.

Reviewer 4 Report

I think the paper is publishable.

The section concerning the limitations of the study can be part of conclusions.

.Author Response
